cognition, neuroscience, behaviour

negative patterning, feature-negative discrimination, associative learning, Pavlovian conditioning, insect, *Drosophila melanogaster*

**Author for correspondence:**
Guillaume Isabel
e-mail: guillaume.isabel@univ-tlse3.fr

# Fruit flies can learn non-elemental olfactory discriminations

Matthias Durrieu[1], Antoine Wystrach[1], Patrick Arrufat[1], Martin Giurfa[1,2,3] and Guillaume Isabel[1]

[1]Research Centre on Animal Cognition, Centre for Integrative Biology, CNRS, University of Toulouse, 118 route de Narbonne, F-31062 Toulouse, France
[2]College of Animal Science (College of Bee Science), Fujian Agriculture and Forestry University, Fuzhou 350002, China
[3]Institut Universitaire de France (IUF), Paris, France

MD, 0000-0003-3675-1140; AW, 0000-0002-3273-7483; MG, 0000-0001-7173-769X; GI, 0000-0003-1312-9826

Associative learning allows animals to establish links between stimuli based on their concomitance. In the case of Pavlovian conditioning, a single stimulus A (the conditional stimulus, CS) is reinforced unambiguously with an unconditional stimulus (US) eliciting an innate response. This conditioning constitutes an 'elemental' association to elicit a learnt response from $A^+$ without US presentation after learning. However, associative learning may involve a 'complex' CS composed of several components. In that case, the compound may predict a different outcome than the components taken separately, leading to ambiguity and requiring the animal to perform so-called non-elemental discrimination. Here, we focus on such a non-elemental task, the negative patterning (NP) problem, and provide the first evidence of NP solving in *Drosophila*. We show that *Drosophila* learn to discriminate a simple component (A or B) associated with electric shocks (+) from an odour mixture composed either partly (called 'feature-negative discrimination' $A^+$ versus $AB^-$) or entirely (called 'NP' $A^+B^+$ versus $AB^-$) of the shock-associated components. Furthermore, we show that conditioning repetition results in a transition from an elemental to a configural representation of the mixture required to solve the NP task, highlighting the cognitive flexibility of *Drosophila*.

## 1. Introduction

The ability to form a link between meaningful events is the cornerstone of associative learning. One of the most studied forms of associative learning is Pavlovian conditioning, where animals learn a conditional relation between a conditioned stimulus (CS) and an unconditioned stimulus (US) [1]. Several protocols can be used to study Pavlovian conditioning. One of them is differential conditioning (DC), which enables the study of how animals learn to associate different CSs with different outcomes (e.g. a reinforced stimulus $A^+$ from a non-reinforced stimulus $B^-$) [2]. In this case, each CS is associated with a distinct US through a simple, non-ambiguous link so that they are learnt independently of each other. In consequence, this type of learning is also referred to as 'elemental learning' [3,4].

However, stimuli in nature may not appear as isolated, distinct elements. Usually, they are compounds constituted of multiple elements. For the elemental learning theory, a compound AB is the linear sum of its elements. In other words, an animal presented with AB would learn the independent relation of A and of B with reinforcement. Yet, several compounds with shared elements and different outcomes could coexist, creating thereby ambiguity at the level of the single elements. This possibility led to the proposal of the configural

(non-elemental) theory, which states that a compound is not learned as the linear sum of its components but as a distinct configuration in which elements would not be fully recognized [5–7]. The validity of this hypothesis has been studied using a discrimination task termed 'negative patterning' (NP) in which a subject has to learn to respond to two single elements reinforced ($A^+$, $B^+$) and to inhibit its response to their non-reinforced compound ($AB^-$). For the elemental learning account, solving this task is impossible as animals reinforced on A and B should respond twice as more to AB. By contrast, for the configural account, NP learning is possible as AB would be perceived as a configuration different from A and B, thus facilitating differentiation [5]. Other studies explored alternative explanations for NP solving and argued that the compound AB would be perceived as the sum of A and B plus a unique cue specific to the AB compound (therefore termed 'unique cue theory') [8]. Within this framework, joint presentation to the animal may result in perceptual interferences such as overshadowing, which suggests that in that case, only part of A and B are perceived within the compound [9]. In either configural or unique cue theory, animals would learn the NP task by focusing on an unambiguous compound-specific cue.

Another task that has received wide attention among learning scholars is the 'feature-negative discrimination' (NF) in which subjects learn to respond to a single element reinforced ($A^+$) but not to a non-reinforced compound $AB^-$ [10]. NF is interesting as it admits both elemental and configural explanations: for the elemental theory, focusing on B (the negative feature) brings an elemental solution to the problem. As stimulus A has an ambiguous valence, being as often reinforced and non-reinforced, B alone suffices as a conditioned inhibitor to respond appropriately. For the non-elemental theory, discrimination is straightforward as the compound AB is a configuration that is unrelated to its elements [10,11].

Studying if and how animals solve the NP and NF discriminations is important to access the mechanisms of associative learning in the animal brain. Given the importance of invertebrates for studies on learning and memory [12,13], research on lobsters [14] and honeybees [15–17] has focused on the capacity of these animals to learn these discrimination problems. In other insect species, other forms of non-elemental learning such as multimodal NF [18], biconditional discrimination [19,20] or contextual learning [21] have been described, yet the solving of NP has remained circumscribed to the honeybee. Attempts to study these phenomena in the fruit fly *Drosophila melanogaster*, an insect model that is used recurrently for studying elemental learning forms [22,23], have failed [24]. This lack of success is regrettable as the fruit fly offers a vast spectrum of neurogenetic tools, which would allow comparing circuits and mechanisms underlying elemental and non-elemental forms of learning. This approach has allowed the olfactory pathways and learning circuitry of the fly to be identified, both from an anatomical (a full connectome of these circuits has been published recently [25]) and from a functional perspective [23]. Robust computational models based on said circuits have been developed and predict that non-elemental learning should be achievable by fruit flies, as normalization events at the perceptual level support the emergence of configural representation of compound stimuli [26]. Thus, studying non-elemental learning from a mechanistic perspective in the fly represents an attractive and accessible goal.

Here, we achieved the first demonstration of the fruit fly's capacity to solve NP and NF in the olfactory domain, thus showing that beyond simple discrimination learning, flies can also solve non-elemental discriminations. In showing this capacity, we determined the associative strategies used by the insects and demonstrated that the processing of olfactory compounds moves along a continuum between elemental and configural processing. Increasing the number of conditioning trials promoted configural processing, enabling flies to solve the NP task.

## 2. Material and methods

### (a) Fly rearing

The wild-type line used in this study was a *Canton-Special* (*Canton-S*) strain. Flies were raised on standard medium at 25°C, approximately 60% humidity and a 12:–12 h light:–dark cycle. The flies were kept in $36 \times 82$ mm plastic tubes containing approximately 20 ml of medium.

### (b) Olfactory conditioning

Odours were diluted in bottles of mineral oil. Odours used were 3-octanol (termed 'A' for the sake of simplicity, 2.27 mM) and 4-methylcyclohexanol ('B', 2.62 mM) (figures 1 and 2). Benzaldehyde ('C', 1.89 mM) was used as a novel odour in some conditions (figure 2). Isoamyl acetate (D) and ethyl butyrate (E) were used as alternative odours (electronic supplementary material, figure S2) to determine if discrimination solving was independent of the pair of odours used. Odours were delivered at the same concentration whether as components (A, B, D or E) or as mixtures (AB or DE). In the case of mixtures, odours were diluted together in the same bottle of mineral oil. All odours and the solvent were from Sigma Aldrich (France). The US consisted of 12 pulses of 1.5 s of 60 V electric shock every 5 s delivered through a metallic grid. Each experiment included two groups of approximately 30 flies (2–4 days old) and was performed using a semi-automated device based on a previous work [27]. In a T-maze, two main phenomena drive the preference of flies towards a compartment or another: on the one hand, the learnt information about the stimuli acquired during conditioning and on the other hand, the fact that odours are rarely completely neutral; at the concentrations used in our work, they are in fact repulsive to naive flies. When two odorants are opposed in the absence of punishment in the T-maze, repulsion balances; yet, if one odorant is opposed to a compound, which is more repulsive, a bias towards the less repulsive stimulus is visible. To disentangle learning from non-learning behavioural components, one of the groups experienced an explicit pairing of CS and US (paired group), while the other group experienced both stimuli unpaired to prevent their association (unpaired group).

### (c) Training

Each training trial consisted of 90 s of acclimatisation, after which flies were subjected to their respective conditioning protocol. Each odour (CS) was presented once for 1 min with an intertrial interval of 1 min. For the paired group (figure 1a), one or two of the olfactory stimulus ($CS^+$) was paired with the US, while the other stimulus ($CS^-$) remained unpunished. In the unpaired group, flies were exposed to 1 min of either shocks or odours, separated by an interval of 1 min. This sequence formed one conditioning cycle.

Flies were subjected to one of three training protocols (figure 1b): a DC in which they had to learn to discriminate a punished from a non-punished odour ($A^+$ versus $B^-$), an NF discrimination in which they had to learn to discriminate a punished odour from a non-punished odour compound ($A^+$ versus

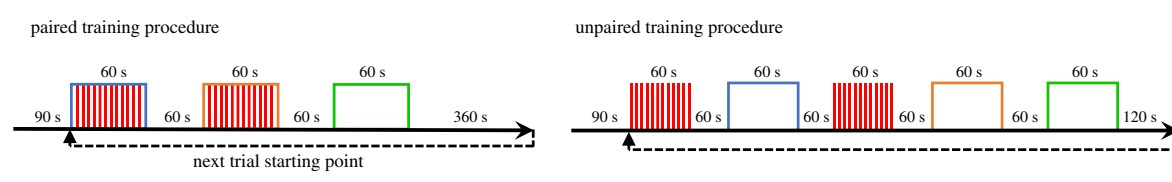

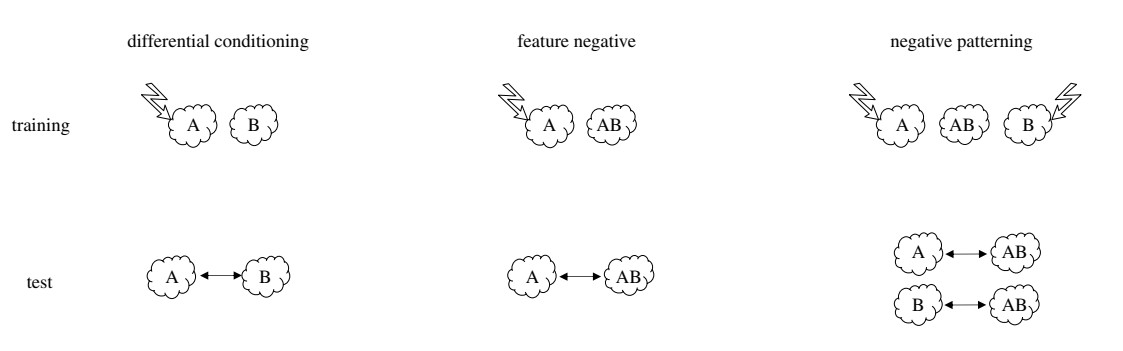

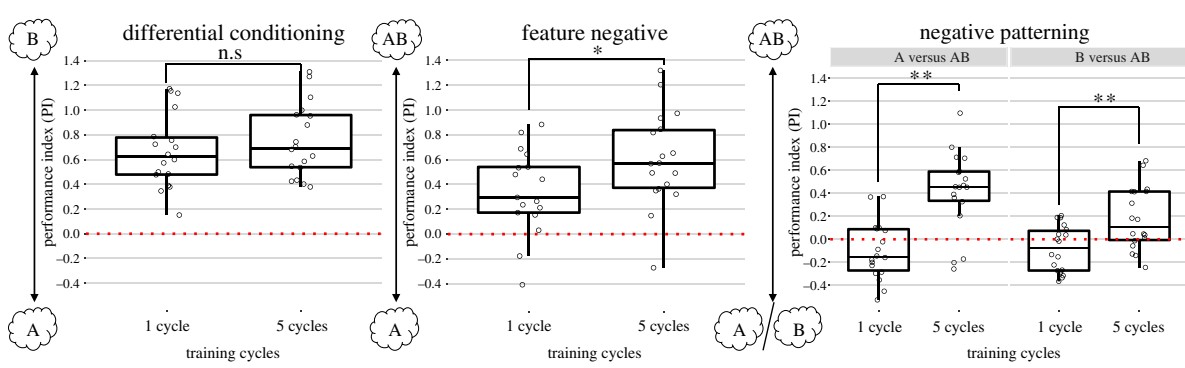

**Figure 1.** (*a*) Schematic of a typical training cycle. Blue and orange boxes show CS presentation, while red bars show US delivery. (*b*) Schematic of the conditioning protocols. Clouds represent the CS odorants while lightning bolts indicate the delivery of electric shock during training. A, 3-octanol; B, 4-methylcyclohexanol. (*c*) Relative PIs computed as the difference between paired and unpaired scores. Performances were compared within the same protocol (i.e. one cycle versus five cycles) but not between protocols. Data are plotted as boxplots. The middle line represents the median, while the upper and lower limits of the box are the 25 and 75% quantiles. The whiskers are the maximum and minimum values of the data that are, respectively, within 1.5 times the interquartile range over the 75th percentile and under the 25th percentile. Raw data are superimposed as jittered dots. 'n.s.' stands for 'non-significant', *p < 0.05, **p < 0.01 after a *t*-test (DC and NF) or after a two-way ANOVA (NP).

$AB^-$) and an NP discrimination in which they had to discriminate two punished odours from a non-punished odour compound ($A^+$, $B^+$ versus $AB^-$). 3-Octanol was always used as the $CS^+$ for the DC and NF protocols (with 4-methylcyclohexanol (DC) and 3-octanol + 4-methylcyclohexanol (NF) as $CS^-$). For the NP protocol, both 3-octanol and 4-methylcyclohexanol were used as $CS^+$ when presented alone and as $CS^-$ when presented as a compound. Thus, flies subjected to DC training faced a pure elemental discrimination. On the contrary, flies trained in the NP protocol could only solve the problem if they adopted a non-elemental strategy. Finally, flies subjected to NF training could solve the problem using either an elemental or a non-elemental strategy. For each protocol, training consisted of either one or five cycles, to enable the study of whether the

amount of experience gathered by flies promoted a particular discriminations strategy.

## (d) Test

After training, flies were transferred to a T-maze [28] where they could choose between the $CS^+$ and the $CS^-$ in the absence of shock during 1 min. In the case of NP, where flies faced two $CS^+$, half of the tests were performed using one of the $CS^+$ while the other half was performed using the other $CS^+$, both presented against the $CS^-$. Flies from paired and unpaired protocols were sequentially tested. At the end of the test, flies in each arm of the T-maze were counted. If paired flies learned the discrimination, they should be mostly located in the CS arm,

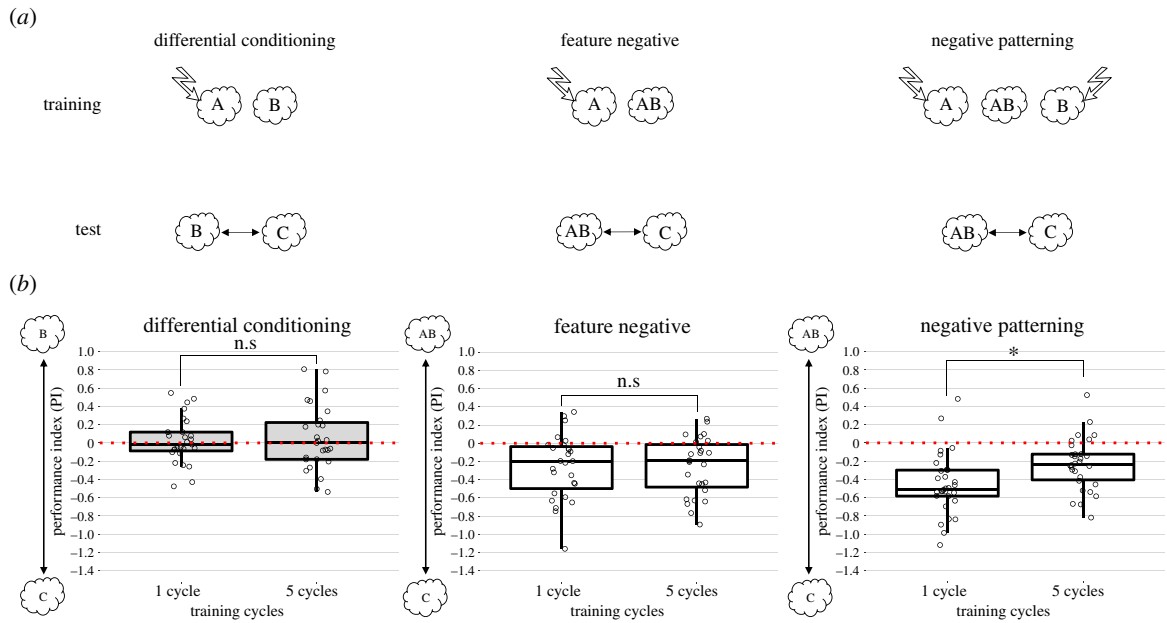

**Figure 2.** (a) Schematic of the tests performed after the three conditioning protocols to determine the nature of the CS representation. A, 3-octanol; B, 4-methyl-cyclohexanol; C, benzaldehyde. (b) Relative PIs computed as the difference between paired and unpaired scores. Performances were compared within the same protocol (i.e. one cycle versus five cycles) but not between protocols. Data are plotted as boxplots. The middle line represents the median. The upper and lower limits of the box are the 25 and 75% quantiles. The whiskers are the maximum and minimum values of the data that are, respectively, within 1.5 times the interquartile range over the 75th percentile and under the 25th percentile. Raw data are superimposed as jittered dots. 'n.s.' stands for 'non-significant', $*p < 0.05$ after a $t$-test. Grey shading indicates performances that were not significantly different from chance level while white filling indicates a significant difference from chance level ($t$-test against zero). (Online version in colour.)

that is, the arm presenting the odour stimulus that was not associated with the shocks during the training. A performance index (PI) was calculated as: (number of flies in the $CS^-$ arm – number of flies in the $CS^+$ arm)/total number of flies. To control for any experimental bias, each replicate consisted of a 'paired group' PI (reflecting associative learning+bias) from which an 'unpaired group' PI (reflecting bias only) was subtracted (electronic supplementary material, figure S1).

### (e) Statistical analysis

PI data were plotted and analysed using R software (3.5.0 v.). Group distributions were tested for normality using the Shapiro–Wilkinson test; homoscedasticity was tested using Bartlett's test. All our experiments met the requirements for parametric statistics. For one or two-sample analyses, we applied one sample or two samples two-tailed Student's tests, respectively. For comparisons involving more than two samples, we used analysis of variance (ANOVA) followed by Tukey's (HSD) *post hoc* tests. A significance level of $p < 0.05$ was set for every experiment.

## 3. Results

Figure 1c (left panel) shows that flies trained in the DC protocol learned the discrimination and preferred the non-punished odour B– to the punished odour A+. Comparing their PIs against 0 by means of a one-sample $t$-test yielded significant differences after both one training cycle ($t = 9.49$, d.f. = 17, $p = 3.33 \times 10^{-8}$) and five training cycles ($t = 10.83$, d.f. = 17, $p = 4{,}76 \times 10^{-9}$). Both PIs did not differ from each other (two-sample $t$-test: $N = 18$ and 18, $t = -0.87$, d.f. = 34, $p = 0.39$), thus providing no evidence for an effect of amount of experience on learned preference.

Flies trained in the NF protocol (figure 1c, middle panel) also learned the discrimination between the single odour punished A+ and the non-punished odour compound AB–. Note

that AB is innately more repulsive than A, but the relative PI shows the expected effect of learning (figure 2b; electronic supplementary material, figure S1). Performance was significantly different from zero after both one cycle of training ($t = 4.13$, d.f. = 16, $p = 7.78 \times 10^{-4}$) or after five cycles ($t = 6.69$, d.f. = 17, $p = 3.77 \times 10^{-6}$). In contrast with DC training, performance improved significantly with the amount of training ($N = 17$ and 18, $t = -2.11$, d.f. = 33, $p = 0.042$). We also tested the flies' performance after the DC or NF protocols balancing odour contingencies, i.e. using B as $CS^+$ and A as $CS^-$ (electronic supplementary material, figure S3).

In the case of flies trained in the NP protocol (figure 1c, right panel), the type of odour used during tests had no significant effect on performance (i.e. 'A versus AB' or 'B versus AB'; two-way ANOVA: $F_{1,69} = 2.72$, $p = 0.10$). Yet, the PI varied significantly depending on the number of training cycles ($F_{1,69} = 34.92$, $p = 1.18 \times 10^{-7}$). After one training cycle, the PI was negative, i.e. biased towards the single odours associated with the shocks during training ($A^+$ or $B^+$) ($t = -2.47$, d.f. = 35, $p = 0.018$). However, after five training cycles, preference was reversed and flies preferred the non-punished compound AB– over the single punished odorants A+, B+ ($t = 5.37$, d.f. = 35, $p = 5.16 \times 10^{-6}$; figure 1c, right panel). These findings show that training repetition is crucial for NP solving as it improved the ability of flies to discriminate the odours with different outcome. To determine if NP solving is independent of the pairs of odours used, we repeated the experiment using different odours (D+, E+ versus DE–; electronic supplementary material, figure S2). The flies also learned the NP discrimination also in this case, showing that the type of odour used during the test phase had no significant effect on performance (i.e. 'D versus DE' or 'E versus DE'; two-way ANOVA: $F_{1,69} = 3.005$, $p = 0.088$). After one trial, flies were not attracted by

the single odours ($t = -0.73$, d.f. = 35, $p = 0.47$). After five trials, they were able to solve the task and showed a significant preference for the compound ($t = 5.21$, d.f. = 35, $p = 8.59 \times 10^{-6}$). This result demonstrates that the ability to solve NP is not specific to the type of odour used in the protocol.

Despite their different complexity, the three protocols assayed opposed one or two CS$^+$ to a CS$^-$. Under these conditions, an animal may learn to avoid the CS$^+$, to be attracted to the CS$^-$, or both. To determine the nature of the associations inculcated by DC, NF and NP, we studied if the CS$^-$ (not reinforced) was learnt as a safe stimulus (inducing attraction), remained neutral (being indifferent to trained animals) or in the case of compounds sharing elements with the reinforced CS$^+$, also gained inhibitory strength, thus eliciting avoidance. To test these options, flies were first trained under the DC, NF or NP protocols (figure 2a) and then given a choice between the CS$^-$ and a novel, neutral odour C. Here again, the preference of trained flies was compared to preference of flies after an unpaired protocol.

After being trained in the DC protocol (figure 2b, left panel), flies tested with the non-punished odour B versus the neutral odour C did not show any preference either after one training cycle ($t = 0.48$, d.f. = 26, $p = 0.64$) or after five training cycles ($t = 0.73$, d.f. = 26, $p = 0.47$). There was no significant difference between PIs corresponding to these different training amounts ($N = 27$ and 27, $t = -0.31$, d.f. = 52, $p = 0.75$). This result thus suggests that the non-punished odour was not perceived as attractive after DC training, and that the number of training cycles had little impact if at all.

After NF conditioning (figure 2b, middle panel), flies significantly preferred the novel odour C to the non-punished compound AB both after one ($t = -3.78$, d.f. = 26, $p = 8.41 \times 10^{-4}$) and five training cycles ($t = -4.15$, d.f. = 26, $p = 3.16 \times 10^{-4}$). This preference was maintained despite the increase in the amount of training ($N = 27$ and 27, $t = -0.03$, d.f. = 52, $p = 0.97$), thus showing that in learning this discrimination, flies assigned an aversive valence to the CS$^-$ despite the fact that it was not paired with shocks. This inhibitory strength must have been acquired via the presence of the punished element A in the compound AB.

Finally, after NP training (figure 2b, left panel), flies also significantly preferred the new odour C to the non-punished compound AB both after one ($t = -7.36$, d.f. = 29, $p = 4.12 \times 10^{-8}$) and five training cycles ($t = -4.46$, d.f. = 29, $p = 1.12 \times 10^{-4}$). Yet, differently from NF training, the flies' aversion for the CS$^-$ was reduced after five training cycles ($N = 30$ and 30, $t = -2.74$, d.f. = 58, $p = 8.10 \times 10^{-3}$), thus showing that increasing the amount of experience diminished the inhibitory strength of the compound. This variation is consistent with a nonlinear processing of the compound, which became less similar to the two single odour components, both being inhibitory after having been paired with shocks.

## 4. Discussion

Our results provide the first evidence of NP solving in an insect other than the honeybee [15,24]. Our results show that flies solved both an NF and an NP discrimination and preferred, in both cases, the odour compound AB that was unpunished during training. In both cases, the compound acquired some inhibitory strength as shown by its avoidance

when confronted with a novel stimulus C, a fact that could reflect an influence of the punished element (A in NF, A and B in NP) in it. Such an influence would be consistent with an elemental processing of the compound (AB = A + B). Yet, in NP, increasing the amount of training diminished this influence, thus promoting a configural processing (AB = new odour).

When interpreted in the light of these theories, our results cannot accommodate a strictly elemental or configural account. Even after five training cycles in NP, the flies still exhibited some avoidance of the CS$^-$ mixture AB, implying that AB$^-$ still carried some of the inhibitory strength associated with its constituents, a result that goes against a purely *configural* compound representation (figure 2b, right panel). In other words, the compound remained repulsive, but was preferred to the 'more repulsive' elements taken alone. This result differs from the response of naive flies, for which a compound is more repulsive than the single odorants, at the concentrations used (electronic supplementary material, figure S1). Our result could thus be interpreted as supporting the 'unique cue theory', which is consistent with observations both in honeybee and *Drosophila*. *In vivo* imaging of the antennal lobes, the primary olfactory centre in the insect brain, suggests that binary mixtures are not coded as the linear sum of their constituents but rather show both similarities and features that are unique to the compound considered [29–32]. However, the fact that the flies' initial choice, which was closer to an elemental representation, was reversed (although not entirely) after NP training shows that increasing training gradually changed the odour representation.

Our results show that the experience of an individual modulates the kind of processing adopted. This modulation is consistent with previous work on honeybees showing, in both the visual and the olfactory modality, that increasing the number of training trials promotes a configural stimulus representation [33,34]. Such a plastic stimulus representation has also been described in other learning paradigms (in crustaceans [14]; larvae [35] and adults *Drosophila* [21,36]; rabbits [37]; humans [38]).

How such a change of odour representation could be implemented in the fly's brain? Firstly, associative learning has been shown to modulate neural representation of odours in the honeybee antennal lobes, a structure involved in olfactory processing, also present in *Drosophila*. Over the course of CS/US associations, odours representation was modified, amplifying the CS$^+$ representation and reducing CS$^+$/CS$^-$ correlation. This could be a first explanatory mechanism for enhanced discrimination abilities necessary for NP resolution [39]. Moreover, pioneer work in the field identified the mushroom bodies as critical structures for the encoding of learnt behaviour in insect brain [40–42]. Interestingly, bees without functional mushroom bodies learn simple olfactory discriminations but are unable to solve NP tasks [43]. Such a failure in NP tasks (as well as in other tasks such as reversal learning) was related to the pharmacological blockade of a specific subset of feedback neurons providing GABAergic signalling to the mushroom bodies [43,44]. The functional equivalent of these neurons in *Drosophila*, the anterior paired lateral (APL) neurons, are equally necessary for reversal learning [45]. Together, this suggests that APL neurons play a crucial role for the modulation of odour representation, and thus for NP solving in flies.

The inhibitory feedback mediated by APL neurons enables a sparse encoding of sensory information by maintaining a low level of activity in the MB, which is needed to discriminate between similar olfactory stimuli [46]. Previous modelling work suggested that normalization at the antennal lobes level is crucial for non-elemental forms of learning as it leads to inhibition of element-specific features, thus enhancing compound-specific (unique cue) salience [26]. APL neurons activity also acts as a gain control mechanism and thus further enhances discrimination at the MB level, which would confirm a hypothesis already proposed in a previous work on NP in the honeybee [43]. Moreover, by modulating its inhibitory activity depending on the task, the APL neurons could mediate a shift along the generalization/discrimination balance as observed here. Indeed, after one cycle of NP conditioning, flies respond to AB as if it were A or B, consistent with odour generalization. Yet, after five cycles, flies change their behaviour and choose AB, even though the second experiment clearly shows that they still perceive the A and B constituents as aversive. This result, in contrast, can be attributed to better discrimination abilities. Interestingly, dopamine signals originating from PPL1 neurons (responsible for aversive US transmission) inhibit APL activity through D2-like receptors, thus modulating olfactory learning [47]. Moreover, APL neurons are activated differently by the $CS^+$ and $CS^-$ [48]. More specifically, the association between the CS and the US results in a diminished APL activity (leading to a reduction in inhibition towards the $CS^+$, facilitating—in principle—its reinforcement). Thus, it is possible that during trials, a differential activation of APL in response to the $CS^+$ versus $CS^-$ produces a form of pruning, leading to the unequivocal reinforcement of the KC associated only with A or B while inhibiting the KC associated both to A/B (the $CS^+$) and to AB (the $CS^-$). If this process is slow compared to the acquisition of the original CS–US association, APL activity would gradually modulate stimulus representation to facilitate the task resolution. In the case of NP, it would help to segregate the representation of AB from those of its elements. APL neurons could thus continually adjust and update representation along a generalization/discrimination continuum based on learning events, besides their role as regular gain control system. Understanding how APL activity might modulate non-elemental olfactory learning constitutes an interesting future research agenda.

So far, olfactory learning in fruit flies was limited to DC protocols, thus reducing the study of associative learning and its neural and molecular mechanisms to its most simple form. Our results thus open new perspectives to explore these mechanisms for higher-order forms of learning and determine if the same or different circuits intervene when different learning strategies are employed.

Data accessibility. Data are available from the Dryad Digital Repository: https://doi.org/10.5061/dryad.v9s4mw6t1 [49].

Authors' contributions. M.D. carried out all the experiments, analysed data, participated in the design of the study and drafted the manuscript; A.W. participated in the design of the study and critically revised the manuscript; P.A. conceived the semi-automated conditioning device; M.G. participated in the design and conceptualization of the study and critically revised the manuscript; G.I. conceived, designed and coordinated the study, and helped draft the manuscript. All authors gave final approval for publication and agree to be held accountable for the work performed therein.

Competing interests. The authors declare no competing interests.

Funding. This work was supported by the French Ministry of Higher Education and Research to M.D. and by the Agence Nationale pour la Recherche (ANR) in the form of the projects entitled funded Toulouse Initiative of Excellence IDEX UNITI (ANR11-IDEX-0002-02) transversality grant and the MoleCulture (ANR-18-CE37-0015) to G.I. M.G. thanks the Institut Universitaire de France for support. G.I. benefited from a CNRS Excellence Chair.

Acknowledgements. We thank Gérard Latil and Audrey Dussutour for providing the necessary material and making the flies food medium; and Stéphane Ferrere for engineering the conditioning set-up.

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
