## [Reviewer comments · Proceedings of the Royal Society B: Biological Sciences]

Review History

RSPB-2020-1234.R0 (Original submission)

Review form: Reviewer 1

Recommendation

Accept with minor revision (please list in comments)

Scientific importance: Is the manuscript an original and important contribution to its field?

Good

General interest: Is the paper of sufficient general interest?

Good

Quality of the paper: Is the overall quality of the paper suitable?

Good

Is the length of the paper justified?

Yes

Should the paper be seen by a specialist statistical reviewer?

No

Do you have any concerns about statistical analyses in this paper? If so, please specify them explicitly in your report.

Yes

It is a condition of publication that authors make their supporting data, code and materials available - either as supplementary material or hosted in an external repository. Please rate, if applicable, the supporting data on the following criteria.

Is it accessible?

No

Is it clear?

No

Is it adequate?

No

Do you have any ethical concerns with this paper?

No

Comments to the Author

This is an interesting paper that shows it is possible to train fruitflies on the non-elemental tasks of negative featural and negative patterning. It slightly overstates the importance of the finding as "Here we provide the first evidence of non-elemental learning in an insect other than honey bee." As far as I am aware it is the first evidence for negative patterning in another insect, but other non-elemental tasks have previously been shown in crickets, moths and flies, see references below.

It is nevertheless of high interest to establish a paradigm for negative patterning in *Drosophila* due to the genetic tools that can then be applied to understand the mechanisms of non-elemental learning. As such, the paper would be strengthened significantly if the experiments had been repeated with a different odour set, to show the general application of the method and result. In fact the methods do not even explicitly state that for the two odours used, the experiment has been repeated (or randomised) with the CS+ and CS- identities swapped.

The paper somewhat conflates the operant definition of non-elemental learning, in which it is observed that the learnt response to a compound stimulus cannot be explained as the sum of responses to its elements, with the potential mechanism, that the compound constitutes a 'distinct entity' in the perceptual system of the animal.

It is an interesting observation, consistent with previous work on honey bees that it takes more training for flies to learn a non-elemental task - note that this itself suggests it is not simply a case of the combined cue appearing to the animal as a 'distinct entity', otherwise it should be just as easily learnt.

The introduction provides a reasonable background to the key questions, but it would be helpful to have a more complete account of configural learning possibilities, in particular, that a compound stimulus could be perceived/learned as 1) the sum of its components (elemental) 2) as the sum of components plus a unique cue arising from the configuration 3) as only partially overlapping with each component plus a unique cue or 4) as a completely unique stimulus. Possibilities 2-4 are glossed over in sentences such as "a compound is not perceived/learned as the linear sum of its components but as a distinct configuration in which elements would not be fully recognized" or "AB would be perceived as a configuration different from A and B, thus facilitating differentiation" or "discrimination is straightforward as the compound AB is a configuration that is unrelated to its elements". There is a substantial difference between the assumption that elements of a compound are "not fully recognised" vs. the assumption that the

compound is "unrelated to its elements". The paper if anything obscures the options outlined above by arguing that their results show that "the processing of olfactory compounds moves along a continuum between elemental and configural processing".

Methods:

- It is mentioned that the odours are innately balanced in repulsion - would it be possible to include a supplementary figure to show this? It is also important that the effects of adaptations to these two odours might differ - here the unpaired results in Supp. Fig. 1 might be sufficient already as showing no preference is induced by exposure alone. In the discussion it is mentioned that the compound is 'twice as repulsive as the single odorants'. This is crucial to interpreting the results so should be more clearly stated in the methods. What would happen if the compound had been balanced (by changing concentrations) against the single odorants?

- Please state explicitly if the same odour was always used as stimulus 'A', and if not, the basis for pooling the results for different odours used as CS+

- Can it be clarified what makes a particular 'unpaired group' a replicate of a particular 'paired group': e.g. did their training occur at the same time, were the flies taken from the same vial...? Otherwise it does not seem statistically justified to treat this as a repeated measures design in which scores are subtracted, as the 'paired' and 'unpaired' groups are not the same flies.

Results

- From the supp fig 1, it would seem the significant learning in the NF procedure is driven almost entirely by the preference in the unpaired group PI, with the flies showing a strong preference for the single odour rather than the compound. The training might then simply reflect an elemental learning in which punishment of the single odour makes it (relatively) less attractive, and the extra cycles are needed (relative to simple discrimination learning) because there is a need to overcome the innate preference.

- The additional testing of the CS- vs. a new odour is useful to gain insight into the learning. For NF, the result appears consistent with the suggestion above, that flies innately find compound odours more aversive than single odours. For NP there does seem to be a change in which the unpunished compound becomes less aversive.

- It is not clear why supp fig 1 does not provide data for NP learning, or for 1 cycle vs. 5 cycles.

Discussion

The claim should be modulated to refer to negative patterning, as other forms of non-elemental discrimination have been shown in other insects:

e.g. biconditional discrimination in crickets and cockroaches:

Matsumoto, Y., & Mizunami, M. (2004). Context-dependent olfactory learning in an insect. *Learning & Memory*, 11(3), 288-293.

Sato, C., Matsumoto, Y., Sakura, M., & Mizunami, M. (2006). Contextual olfactory learning in cockroaches. *Neuroreport*, 17(5), 553-557.

Negative Feature discrimination (although as noted in the current paper this can potentially be explained with elemental learning) in moths:

Balkenius, A., & Hansson, B. (2012). Discrimination training with multimodal stimuli changes activity in the mushroom body of the hawkmoth *Manduca sexta*. *PLoS One*, 7(4), e32133.

In fruitflies, context dependent learning, which can be equivalent to NF or biconditional learning, depending on the paradigm, has been previously demonstrated:

Brembs, B., & Wiener, J. (2006). Context and occasion setting in *Drosophila* visual learning. *Learning & Memory*, 13(5), 618-628.

This is a few examples from a fairly brief search of the literature - the authors should make a more comprehensive survey to contextualise their claims.

As outlined in the general comments above, it is not a good representation of the extensive literature on configural learning to suggest that current researchers might believe there to be a "strict dichotomy between an elemental and a configural representation of odours" and that it is thus a contribution of this paper to have shown a "continuum" of processing.

It is appropriate to consider the MB architecture as a plausible substrate for non-elemental learning. But the discussion could be more mechanistic. There are two ways in which the KC pattern resulting from presentation of two odours together could differ from the sum of each odour individually, even if the activation at the PN level is a simple sum. Firstly, KC seem to act as coincidence detectors for their PN inputs; there may be some KCs that have inputs from PNs that respond to different odours; hence are only activated when both odours are present. Second, if inhibitory feedback limits the total number of active KCs, then even if the initial activation of KCs by PNs for the compound was an exact overlap of the response to the elements, the inhibition would 'prune' this representation to remove any more weakly responding KCs, hence the resulting activation would again represent the compound differently from the sum of the elements. These mechanisms have been explored in some of the modelling papers cited, e.g. 19, and in actual recordings from *Drosophila* KC so deserve clearer presentation,

Review form: Reviewer 2

Recommendation

Accept with minor revision (please list in comments)

Scientific importance: Is the manuscript an original and important contribution to its field?

Excellent

General interest: Is the paper of sufficient general interest?

Excellent

Quality of the paper: Is the overall quality of the paper suitable?

Excellent

Is the length of the paper justified?

Yes

Should the paper be seen by a specialist statistical reviewer?

No

Do you have any concerns about statistical analyses in this paper? If so, please specify them explicitly in your report.

No

It is a condition of publication that authors make their supporting data, code and materials available - either as supplementary material or hosted in an external repository. Please rate, if applicable, the supporting data on the following criteria.

Is it accessible?

No

Is it clear?

No

Is it adequate?

No

Do you have any ethical concerns with this paper?

No

Comments to the Author

Durrieu et al present the first behavioral evidence that *Drosophila* can learn “non-elemental” conditioning tasks, e.g., A and B are paired with shock when presented individually, but the mixture AB is not paired with shock, thus the fly goes to AB instead of A or B.

This is an important finding because it is clearly predicted by our current understanding of the *Drosophila* olfactory system, yet it has never been observed (indeed, a previous study tested for non-elemental learning and did not see it). Thus, this study provides important experimental support for current theoretical understandings of olfactory sensory coding in *Drosophila*. I support publication with minor revisions.

1. Discussion: “This result differs from the response of naïve flies, for which a compound is twice as repulsive as the single odorants, at the concentrations used.” - Is this derived from the unpaired PI in supplementary figure 1, where untrained flies prefer A over AB? The authors should explain more explicitly how the figure “twice as repulsive” is calculated, or if they aren’t confident in the quantitative estimate, soften it to a qualitative “more repulsive.”
2. Discussion: “By modulating its inhibitory activity, the APL neurons could in theory mediate a shift along the generalization / discrimination balance as observed here.” I don’t think the authors have shown a shift between generalization and discrimination - can they clarify what they mean here, and provide a hypothesis for how APL might be involved?
3. Can the authors speculate on why, for negative patterning, flies prefer A over AB after 1 training cycle? Why is repetition required to get configural learning? Is it simply that configural learning is difficult? Or would repetitive training shift a balance between elemental vs. non-elemental learning? Is there any theoretical reason to expect such a shift?
4. Supplementary figure 1 - can the authors show similar data for the negative patterning experiments as well?
5. The authors should clarify how AB mixtures were generated - were the two odors at the same concentration as when presented singly (so that an MCH-octanol mixture would have 2.27 mM octanol and 2.62 mM MCH) or were they at half the ‘single-odor’ concentration? [The latter would complicate the interpretation of the negative featuring experiment - the flies might not be seeing AB as a different ‘configuration’ from A but rather as a lower concentration of A.] Were the mixtures “liquid” mixtures (i.e. MCH and octanol diluted in the same volume of mineral oil) or mixtures of 2 separate odor-saturated air streams?

2 points that are only suggestions:

6. The authors may wish to frame the paper more around the puzzle that non-elemental learning hasn't been observed even though is clearly predicted from the field's current understanding of olfactory sensory coding in flies (Wessnitzer et al - but even intuitively from first principles - because the KC population response to AB probably includes KCs that didn't respond to either A or B, thanks to normalization and lateral interactions between glomeruli in the antennal lobe). They do mention this problem but I am suggesting to make the tension between experiment and theory more central to the Intro and Abstract. It might make the paper more visible to the sensory coding field (not just the behavior field).

7. Given that Wessnitzer et al's model can only learn negative patterning with normalization in the antennal lobe, it may be worth discussing that being able to learn negative patterning might be one functional 'purpose' for having normalization (on top of the usual gain control reasons).

Minor points

Discussion: "Even after five training cycles in NP, the flies still exhibited some avoidance of the CS- mixture AB, implying that AB- still carried some of the inhibitory strength associated with its constituents, and thus refutes a purely configural representation (figure 2b, left panel)." - should this be figure 2b, *right* panel?

Typo in 3rd-to-last para: "How such a change in of odour representation could be implemented in the fly's brain?"

Decision letter (RSPB-2020-1234.R0)

02-Jul-2020

Dear Dr Isabel:

Your manuscript has now been peer reviewed and the reviews have been assessed by an Associate Editor. The reviewers' comments (not including confidential comments to the Editor) and the comments from the Associate Editor are included at the end of this email for your reference. As you will see, the reviewers and the Editors have raised some concerns with your manuscript and we would like to invite you to revise your manuscript to address them.

Research ethics:

Use of animals and field studies:

Please submit a copy of your revised paper within three weeks. If we do not hear from you within this time your manuscript will be rejected. If you are unable to meet this deadline please let us know as soon as possible, as we may be able to grant a short extension.

Best wishes,
Dr Robert Barton
mailto: proceedingsb@royalsociety.org

Associate Editor
Comments to Author:

Your manuscript has now been reviewed by two experts in the field who both agree the study has good scientific importance, is well done and of broad interest. Both reviewers make several constructive suggestions on how to clarify and strengthen the manuscript. In addition, please note that it is the journal's policy that all data, code and materials must be publicly available, either by inclusion in a supplementary material or submitted to an online repository. Currently this is not the case. In the revised submission the authors must include this information, including data from all individual trials, such that the analyses could be repeated as described. The manuscript cannot be accepted without this information.

Reviewer(s)' Comments to Author:

Referee: 1

Comments to the Author(s)

This is an interesting paper that shows it is possible to train fruitflies on the non-elemental tasks of negative featural and negative patterning. It slightly overstates the importance of the finding as "Here we provide the first evidence of non-elemental learning in an insect other than honey bee." As far as I am aware it is the first evidence for negative patterning in another insect, but other non-elemental tasks have previously been shown in crickets, moths and flies, see references below.

It is nevertheless of high interest to establish a paradigm for negative patterning in *Drosophila* due to the genetic tools that can then be applied to understand the mechanisms of non-elemental learning. As such, the paper would be strengthened significantly if the experiments had been repeated with a different odour set, to show the general application of the method and result. In fact the methods do not even explicitly state that for the two odours used, the experiment has been repeated (or randomised) with the CS+ and CS- identities swapped.

The paper somewhat conflates the operant definition of non-elemental learning, in which it is observed that the learnt response to a compound stimulus cannot be explained as the sum of responses to its elements, with the potential mechanism, that the compound constitutes a 'distinct entity' in the perceptual system of the animal.

It is an interesting observation, consistent with previous work on honey bees that it takes more training for flies to learn a non-elemental task - note that this itself suggests it is not simply a case of the combined cue appearing to the animal as a 'distinct entity', otherwise it should be just as easily learnt.

The introduction provides a reasonable background to the key questions, but it would be helpful to have a more complete account of configural learning possibilities, in particular, that a compound stimulus could be perceived/learned as 1) the sum of its components (elemental) 2) as the sum of components plus a unique cue arising from the configuration 3) as only partially overlapping with each component plus a unique cue or 4) as a completely unique stimulus. Possibilities 2-4 are glossed over in sentences such as "a compound is not perceived/learned as the linear sum of its components but as a distinct configuration in which elements would not be fully recognized" or "AB would be perceived as a configuration different from A and B, thus facilitating differentiation" or "discrimination is straightforward as the compound AB is a configuration that is unrelated to its elements". There is a substantial difference between the assumption that elements of a compound are "not fully recognised" vs. the assumption that the

compound is "unrelated to its elements". The paper if anything obscures the options outlined above by arguing that their results show that "the processing of olfactory compounds moves along a continuum between elemental and configural processing".

Methods:

- It is mentioned that the odours are innately balanced in repulsion - would it be possible to include a supplementary figure to show this? It is also important that the effects of adaptations to these two odours might differ - here the unpaired results in Supp. Fig. 1 might be sufficient already as showing no preference is induced by exposure alone. In the discussion it is mentioned that the compound is 'twice as repulsive as the single odorants'. This is crucial to interpreting the results so should be more clearly stated in the methods. What would happen if the compound had been balanced (by changing concentrations) against the single odorants?

- Please state explicitly if the same odour was always used as stimulus 'A', and if not, the basis for pooling the results for different odours used as CS+

- Can it be clarified what makes a particular 'unpaired group' a replicate of a particular 'paired group': e.g. did their training occur at the same time, were the flies taken from the same vial...? Otherwise it does not seem statistically justified to treat this as a repeated measures design in which scores are subtracted, as the 'paired' and 'unpaired' groups are not the same flies.

Results

- From the supp fig 1, it would seem the significant learning in the NF procedure is driven almost entirely by the preference in the unpaired group PI, with the flies showing a strong preference for the single odour rather than the compound. The training might then simply reflect an elemental learning in which punishment of the single odour makes it (relatively) less attractive, and the extra cycles are needed (relative to simple discrimination learning) because there is a need to overcome the innate preference.

- The additional testing of the CS- vs. a new odour is useful to gain insight into the learning. For NF, the result appears consistent with the suggestion above, that flies innately find compound odours more aversive than single odours. For NP there does seem to be a change in which the unpunished compound becomes less aversive.

- It is not clear why supp fig 1 does not provide data for NP learning, or for 1 cycle vs. 5 cycles.

Discussion

The claim should be modulated to refer to negative patterning, as other forms of non-elemental discrimination have been shown in other insects:

e.g. biconditional discrimination in crickets and cockroaches:

Matsumoto, Y., & Mizunami, M. (2004). Context-dependent olfactory learning in an insect. *Learning & Memory*, 11(3), 288-293.

Sato, C., Matsumoto, Y., Sakura, M., & Mizunami, M. (2006). Contextual olfactory learning in cockroaches. *Neuroreport*, 17(5), 553-557.

Negative Feature discrimination (although as noted in the current paper this can potentially be explained with elemental learning) in moths:

Balkenius, A., & Hansson, B. (2012). Discrimination training with multimodal stimuli changes activity in the mushroom body of the hawkmoth *Manduca sexta*. *PLoS One*, 7(4), e32133.

In fruitflies, context dependent learning, which can be equivalent to NF or biconditional learning, depending on the paradigm, has been previously demonstrated:

Brembs, B., & Wiener, J. (2006). Context and occasion setting in *Drosophila* visual learning. *Learning & Memory*, 13(5), 618-628.

This is a few examples from a fairly brief search of the literature - the authors should make a more comprehensive survey to contextualise their claims.

As outlined in the general comments above, it is not a good representation of the extensive literature on configural learning to suggest that current researchers might believe there to be a "strict dichotomy between an elemental and a configural representation of odours" and that it is thus a contribution of this paper to have shown a "continuum" of processing.

It is appropriate to consider the MB architecture as a plausible substrate for non-elemental learning. But the discussion could be more mechanistic. There are two ways in which the KC pattern resulting from presentation of two odours together could differ from the sum of each odour individually, even if the activation at the PN level is a simple sum. Firstly, KC seem to act as coincidence detectors for their PN inputs; there may be some KCs that have inputs from PNs that respond to different odours; hence are only activated when both odours are present. Second, if inhibitory feedback limits the total number of active KCs, then even if the initial activation of KCs by PNs for the compound was an exact overlap of the response to the elements, the inhibition would 'prune' this representation to remove any more weakly responding KCs, hence the resulting activation would again represent the compound differently from the sum of the elements. These mechanisms have been explored in some of the modelling papers cited, e.g. 19, and in actual recordings from *Drosophila* KC so deserve clearer presentation,

Referee: 2

Comments to the Author(s)

Durrieu et al present the first behavioral evidence that *Drosophila* can learn "non-elemental" conditioning tasks, e.g., A and B are paired with shock when presented individually, but the mixture AB is not paired with shock, thus the fly goes to AB instead of A or B.

This is an important finding because it is clearly predicted by our current understanding of the *Drosophila* olfactory system, yet it has never been observed (indeed, a previous study tested for non-elemental learning and did not see it). Thus, this study provides important experimental support for current theoretical understandings of olfactory sensory coding in *Drosophila*. I support publication with minor revisions.

1. Discussion: "This result differs from the response of naïve flies, for which a compound is twice as repulsive as the single odorants, at the concentrations used." - Is this derived from the unpaired PI in supplementary figure 1, where untrained flies prefer A over AB? The authors should explain more explicitly how the figure "twice as repulsive" is calculated, or if they aren't confident in the quantitative estimate, soften it to a qualitative "more repulsive."
2. Discussion: "By modulating its inhibitory activity, the APL neurons could in theory mediate a shift along the generalization / discrimination balance as observed here." I don't think the authors have shown a shift between generalization and discrimination - can they clarify what they mean here, and provide a hypothesis for how APL might be involved?
3. Can the authors speculate on why, for negative patterning, flies prefer A over AB after 1 training cycle? Why is repetition required to get configural learning? Is it simply that configural learning is difficult? Or would repetitive training shift a balance between elemental vs. non-elemental learning? Is there any theoretical reason to expect such a shift?

4. Supplementary figure 1 - can the authors show similar data for the negative patterning experiments as well?

5. The authors should clarify how AB mixtures were generated - were the two odors at the same concentration as when presented singly (so that an MCH-octanol mixture would have 2.27 mM octanol and 2.62 mM MCH) or were they at half the 'single-odor' concentration? [The latter would complicate the interpretation of the negative featuring experiment - the flies might not be seeing AB as a different 'configuration' from A but rather as a lower concentration of A.] Were the mixtures "liquid" mixtures (i.e. MCH and octanol diluted in the same volume of mineral oil) or mixtures of 2 separate odor-saturated air streams?

2 points that are only suggestions:

6. The authors may wish to frame the paper more around the puzzle that non-elemental learning hasn't been observed even though is clearly predicted from the field's current understanding of olfactory sensory coding in flies (Wessnitzer et al - but even intuitively from first principles - because the KC population response to AB probably includes KCs that didn't respond to either A or B, thanks to normalization and lateral interactions between glomeruli in the antennal lobe). They do mention this problem but I am suggesting to make the tension between experiment and theory more central to the Intro and Abstract. It might make the paper more visible to the sensory coding field (not just the behavior field).

7. Given that Wessnitzer et al's model can only learn negative patterning with normalization in the antennal lobe, it may be worth discussing that being able to learn negative patterning might be one functional 'purpose' for having normalization (on top of the usual gain control reasons).

Minor points

Discussion: "Even after five training cycles in NP, the flies still exhibited some avoidance of the CS- mixture AB, implying that AB- still carried some of the inhibitory strength associated with its constituents, and thus refutes a purely configural representation (figure 2b, left panel)." - should this be figure 2b, *right* panel?

Typo in 3rd-to-last para: "How such a change in of odour representation could be implemented in the fly's brain?"

Author's Response to Decision Letter for (RSPB-2020-1234.R0)

See Appendix A.

RSPB-2020-1234.R1 (Revision)

Review form: Reviewer 1

Recommendation

Accept with minor revision (please list in comments)

Scientific importance: Is the manuscript an original and important contribution to its field?

Good

General interest: Is the paper of sufficient general interest?

Good

Quality of the paper: Is the overall quality of the paper suitable?

Good

Is the length of the paper justified?

Yes

Should the paper be seen by a specialist statistical reviewer?

No

Do you have any concerns about statistical analyses in this paper? If so, please specify them explicitly in your report.

Yes

It is a condition of publication that authors make their supporting data, code and materials available - either as supplementary material or hosted in an external repository. Please rate, if applicable, the supporting data on the following criteria.

Is it accessible?

Yes

Is it clear?

Yes

Is it adequate?

Yes

Do you have any ethical concerns with this paper?

No

Comments to the Author

This revision has addressed most of my concerns with the original paper. In particular the authors have significantly clarified i) which aspect of their results are original, ii) the mechanistic implications of non-elemental learning, related to known neural circuitry and ii) the generalisability of the results, by repeating the key experiments with alternative odour combinations. Overall, this provides a convincing story that should be of wide interest.

I have the following minor concerns:

lines 173-175 It is not appropriate to conclude that H0 is supported "showing that the amount of experience of flies did not affect the learning of this elemental discrimination" from a non-significant statistical difference; rather, the experiment did not provide evidence for an effect of amount of experience.

lines 155-157 Although I accept it is a common procedure in this field, I am still not wholly convinced by the assumption that sets of paired and unpaired groups can be considered true replicates and hence have their scores subtracted to obtain 'corrected PI' scores for analysis. Although the same mean value is obtained by using the 'corrected PI' or the difference of the mean Paired and mean Unpaired, the variance calculation that forms the basis of the t-test is different under these two procedures: what is effectively a paired t-test will give a different result to a two-sample t-test with independent groups. Moreover, it seems very relevant to the discussion of mechanisms that the bias in the 'corrected' PI is sometimes a result of biased behaviour in the unpaired rather than the paired group (see below). As such, the terminology 'corrected PI' seems misleading to me.

I do not fully understand their response to my suggestion, in my review, that the NF learning could be elemental. I did not suggest that the single odour 'A' was attractive, but that it is 'relatively' attractive (i.e. less unattractive) compared to the compound 'AB', and trial repetition makes A more and more aversive (rather than AB less and less aversive) leading to a 'balance' of aversion for the paired group, a remaining imbalance in the unpaired group, and hence a significant change in the 'corrected' PI index. However I accept that this account still implies that AB is processed differently to A+B (i.e. non-elementally) as otherwise AB aversiveness should change in step with the change in aversiveness of A. However, it feels to me that the revision of the paper rather 'buries' the relevant fact that compounds are innately more aversive, which is relevant to understanding the mechanisms.

Supp fig 1, it is good to have included NP data, but I would still like, for completeness, to see the same data for 1 and 5 cycles, as I believe it is important to understand whether or not changes in behaviour are seen during unpaired training in these paradigms.

Review form: Reviewer 2

Recommendation

Accept as is

Scientific importance: Is the manuscript an original and important contribution to its field?

Excellent

General interest: Is the paper of sufficient general interest?

Excellent

Quality of the paper: Is the overall quality of the paper suitable?

Excellent

Is the length of the paper justified?

Yes

Should the paper be seen by a specialist statistical reviewer?

No

Do you have any concerns about statistical analyses in this paper? If so, please specify them explicitly in your report.

No

It is a condition of publication that authors make their supporting data, code and materials available - either as supplementary material or hosted in an external repository. Please rate, if applicable, the supporting data on the following criteria.

Is it accessible?

Yes

Is it clear?

Yes

Is it adequate?

Yes

Do you have any ethical concerns with this paper?

No

Comments to the Author

My comments have been fully addressed.

Ref 48 is missing the journal and part of the title but this can be corrected in the proof.

Decision letter (RSPB-2020-1234.R1)

13-Oct-2020

Dear Dr Isabel

I am pleased to inform you that your manuscript RSPB-2020-1234.R1 entitled "Fruit flies can learn non-elemental olfactory discriminations" has been accepted for publication in Proceedings B.

The referee(s) have recommended publication, but also suggest some minor revisions to your manuscript. Therefore, I invite you to respond to the referee(s)' comments and revise your manuscript. Because the schedule for publication is very tight, it is a condition of publication that you submit the revised version of your manuscript within 7 days. If you do not think you will be able to meet this date please let us know.

Sincerely,

Dr Robert Barton

Associate Editor:

Board Member: 1

Author's Response to Decision Letter for (RSPB-2020-1234.R1)

See Appendix B.

Decision letter (RSPB-2020-1234.R2)

19-Oct-2020

Dear Dr Isabel

I am pleased to inform you that your manuscript entitled "Fruit flies can learn non-elemental olfactory discriminations" has been accepted for publication in Proceedings B.

Your article has been estimated as being 9 pages long. Our Production Office will be able to confirm the exact length at proof stage.

Open Access

Paper charges

Sincerely,

Proceedings B

Appendix A

C B I
Centre de Biologie
Intégrative - Toulouse

Centre de Recherches sur la Cognition Animale-Centre de Biologie Intégrative
Université Paul Sabatier - CNRS, Bât IVR 3
118 Route de Narbonne, 31062 TOULOUSE Cedex 4, FRANCE
<http://crca.cbi-toulouse.fr/>

Corresponding author : Dr. Guillaume Isabel
guillaume.isabel@univ-tlse3.fr

E-mail :

Dear Editor,

Please find a revised version of our manuscript entitled "***Fruit flies can learn non-elemental olfactory discriminations***" (RSPB-2020-1234).

We would like to thank the editor for enabling our paper to be reviewed. We also thank the referees for their insightful comments, which were considered to correct and strengthen our paper. This new version carefully accounts for all your comments and those of the referee, in a way that we describe below, with comments in black and our replies in red. We hope that you will find this new version satisfactory.

All the best,

Guillaume Isabel (PhD) on behalf of all co-authors.

Reviewer(s)' Comments to Author:

Referee: 1

Comments to the Author(s)

This is an interesting paper that shows it is possible to train fruitflies on the non-elemental tasks of negative featural and negative patterning. It slightly overstates the importance of the finding as "Here we provide the first evidence of non-elemental learning in an insect other than honey bee." As far as I am aware it is the first evidence for negative patterning in another insect, but other non-elemental tasks have previously been shown in crickets, moths and flies, see references below.

We thank the reviewer for this useful comment. Indeed, different forms of contextual learning and occasion setting, demonstrated in other insects, can be considered as cases of non-elemental learning. We should

CRCA – CBI – CNRS Université Paul Sabatier
118 route de Narbonne – Bât 4R3B3
31062 Toulouse Cedex 4
France

C B I
Centre de Biologie
Intégrative - Toulouse

UNIVERSITÉ
TOULOUSE III
PAUL SABATIER
Université
de Toulouse

have probably specified that what he had in mind is the particular case of patterning experiments (i.e. positive or negative patterning, PP and NP, respectively). These discrimination problems constitute a whole area of research *per se* (see for instance the literature on NP and PP in vertebrates and the role of the hippocampus). Moreover, the interest of our work refers to previous reports stating that fruit flies were unable to solve this discrimination (Young et al., 2011, ref 24). We have lowered our claim and specified that we focused on patterning discriminations, which have been rarely addressed in insects beyond the honey bee. We have also cited examples of non-elemental tasks in other insects.

It is nevertheless of high interest to establish a paradigm for negative patterning in *Drosophila* due to the genetic tools that can then be applied to understand the mechanisms of non-elemental learning. As such, the paper would be strengthened significantly if the experiments had been repeated with a different odour set, to show the general application of the method and result.

We fully agree with this statement. We have therefore followed the Reviewer's advice and performed additional experiments to test NP abilities using two additional odours : IsoamylAcetate and EthylButyrate [also used in some papers on *Drosophila* learning and memory (Lin *et al.*, 2014) ref 45]. Our new results demonstrate that NP is not only observable in the case of 3-Octanol and 4-Methylcyclohexanol, but extends to the additional pair of odours . The new NP results with IsoamylAcetate and EthylButyrate are shown in figure S2.

In fact the methods do not even explicitly state that for the two odours used, the experiment has been repeated (or randomised) with the CS+ and CS- identities swapped.

Regarding NP, it is not possible to perform the suggested swapping given that the single odors are the CS+ and the compound the CS- (A+ B+ vs. AB-). This ambiguity is precisely what makes the interest of the discrimination problem. Should we swap the valence of single odors and compound, we would be then studying another problem, which is the positive patterning discrimination. That was not our goal.

Regarding Differential Conditioning and Negative Featural, we performed the corresponding experiments in which contingencies were reversed to fulfill the Reviewer's request. Our results show that performance is independent of this factor, an important point for which we thank the Reviewer (see figure S3).

The paper somewhat conflates the operant definition of non-elemental learning, in which it is observed that the learnt response to a compound stimulus cannot be explained as the sum of responses to its elements, with the potential mechanism, that the compound constitutes a 'distinct entity' in the perceptual system of the animal.

We agree with the referee, and we now made sure our manuscript no longer displays this ambiguity. This distinction is usually swamped in the literature, where 'non-elemental' is equally used to describe the protocol and the actual mechanism, so-called 'animal representation'. Here we show that the elemental/non-elemental distinction cannot account for the actual animal mechanisms as our flies observed behaviour show typical features both from elemental and non-elemental representation, which can however be well explained regarding the neural circuitry. Therefore, it becomes evident that non-elemental should be used only as an operant definition, reflecting the experimental task, and should not refer to the animal's perceptual system. We hope that the our new version makes this point clearer.

C B I
Centre de Biologie
Intégrative - Toulouse

It is an interesting observation, consistent with previous work on honey bees that it takes more training for flies to learn a non-elemental task - note that this itself suggests it is not simply a case of the combined cue appearing to the animal as a 'distinct entity', otherwise it should be just as easily learnt.

Indeed, thanks.

The introduction provides a reasonable background to the key questions, but it would be helpful to have a more complete account of configural learning possibilities, in particular, that a compound stimulus could be perceived/learned as 1) the sum of its components (elemental) 2) as the sum of components plus a unique cue arising from the configuration 3) as only partially overlapping with each component plus a unique cue or 4) as a completely unique stimulus. Possibilities 2-4 are glossed over in sentences such as "a compound is not perceived/learned as the linear sum of its components but as a distinct configuration in which elements would not be fully recognized" or "AB would be perceived as a configuration different from A and B, thus facilitating differentiation" or "discrimination is straightforward as the compound AB is a configuration that is unrelated to its elements". There is a substantial difference between the assumption that elements of a compound are "not fully recognised" vs. the assumption that the compound is "unrelated to its elements". The paper if anything obscures the options outlined above by arguing that their results show that "the processing of olfactory compounds moves along a continuum between elemental and configural processing".

We understand the expressed concerns. For the sake of clarity, we reworked this part (Lines 54-60). However, because our stance is that it is healthier to interpret the potential mechanisms directly in the light of neural circuits rather than the aforementioned notions (1 to 4), we would stick to such a neural interpretation in the discussion.

Methods:

- It is mentioned that the odours are innately balanced in repulsion - would it be possible to include a supplementary figure to show this? It is also important that the effects of adaptations to these two odours might differ - here the unpaired results in Supp. Fig. 1 might be sufficient already as showing no preference is induced by exposure alone. In the discussion it is mentioned that the compound is 'twice as repulsive as the single odorants'. This is crucial to interpreting the results so should be more clearly stated in the methods. What would happen if the compound had been balanced (by changing concentrations) against the single odorants?

As mentioned by the referee, the unpaired group in figure S1 demonstrates no preference after exposure in the DC condition, which suggests that in absence of associative learning, the odours (at the concentrations used) were balanced, i.e. equally chosen.

Regarding the influence of odour concentrations on Negative Patterning resolution, this constitute a question per se that was out of the scope of our work. Nonetheless, this question has been addressed in honeybee, where three CS+/CS- concentration ratios have been used (1:1, 1:2 and 1:3) (Deisig et al., 2001, ref 15). In all cases, the bees solved the task without any significant concentration effect.

C B I
Centre de Biologie
Intégrative - Toulouse

Concerning the repulsive nature of the compound, we undertoned our statement (Line 117). In the case of NF, we show in figure S1 that the simple exposure to A vs AB leads to a significant bias towards A, which we interpret as AB being more repulsive than A.

- Please state explicitly if the same odour was always used as stimulus 'A', and if not, the basis for pooling the results for different odours used as CS+

3-Octanol was always used as odor A, yet we provided supplementary results with different odours. To avoid confusions, we gave them different names.

- Can it be clarified what makes a particular 'unpaired group' a replicate of a particular 'paired group': e.g. did their training occur at the same time, were the flies taken from the same vial...? Otherwise it does not seem statistically justified to treat this as a repeated measures design in which scores are subtracted, as the 'paired' and 'unpaired' groups are not the same flies.

Paired and Unpaired flies come from the same vials and have the same age. They are conditioned in parallel, with conditioning protocols designed to begin and end at the same time. Finally, during test phase, Paired and Unpaired groups are interspersed and subsequently tested using the same T-Maze. Incidentally, computing the mean performance index or expressing it as the difference between the mean Paired score and Unpaired score is mathematically equal. Finally, *Drosophila* field has been traditionally studying learning and memory using a slightly different protocol that is characterized by a similar calculation method (A+ B- and B+ A- performances are averaged in pairs after being tested in succession, see Tully & Quinn, 1985, Ref 28).

Results

- From the supp fig 1, it would seem the significant learning in the NF procedure is driven almost entirely by the preference in the unpaired group PI, with the flies showing a strong preference for the single odour rather than the compound. The training might then simply reflect an elemental learning in which punishment of the single odour makes it (relatively) less attractive, and the extra cycles are needed (relative to simple discrimination learning) because there is a need to overcome the innate preference.

- The additional testing of the CS- vs. a new odour is useful to gain insight into the learning. For NF, the result appears consistent with the suggestion above, that flies innately find compound odours more aversive than single odours. For NP there does seem to be a change in which the unpunished compound becomes less aversive.

The Reviewer's interpretation assumes that 3-Octanol (alone) is innately attractive. However, this is not the case. In fact, naïve flies avoid 3-Octanol when presented against the solvent (see Plaçais et al., 2012).

This point is further strengthened by the results observed in the Unpaired condition of Differential Conditioning in the supp fig1: When presented vs B (which has been diluted so that the innate avoidance towards A and B cancel each other), unpaired flies choose equally between A and B, which again shows that odour A is not attractive.

Nonetheless, in protocols involving compounds, the unpaired group clearly shows an innate repulsion towards AB. Previous work in *Drosophila* (Tully & Quinn, 1985) has shown that 1 trial (as used in our work)

C B I
Centre de Biologie
Intégrative - Toulouse

is enough for flies to acquire a robust avoidance of the CS+ (3-Octanol). If anything, this result suggests that trial repetition changes AB representation to be less and less aversive compared to A / B.

It is, however, true that a recent paper demonstrated a remarkable symmetry between aversive and appetitive pathways (Felsenberg et al., 2018), which could indeed lead to think that less attractive and more aversive are one and the same, thus making the distinction contrived.

- It is not clear why supp fig 1 does not provide data for NP learning, or for 1 cycle vs. 5 cycles.

Supp Fig1 is only a proof of concept regarding the paired/unpaired procedure. Yet we added the NP data in order to avoid any misunderstanding.

Discussion

The claim should be modulated to refer to negative patterning, as other forms of non-elemental discrimination have been shown in other insects:

e.g. biconditional discrimination in crickets and cockroaches:

Matsumoto, Y., & Mizunami, M. (2004). Context-dependent olfactory learning in an insect. *Learning & Memory*, 11(3), 288-293.

Sato, C., Matsumoto, Y., Sakura, M., & Mizunami, M. (2006). Contextual olfactory learning in cockroaches. *Neuroreport*, 17(5), 553-557.

Negative Feature discrimination (although as noted in the current paper this can potentially be explained with elemental learning) in moths:

Balkenius, A., & Hansson, B. (2012). Discrimination training with multimodal stimuli changes activity in the mushroom body of the hawkmoth *Manduca sexta*. *PLoS One*, 7(4), e32133.

In fruitflies, context dependent learning, which can be equivalent to NF or biconditional learning, depending on the paradigm, has been previously demonstrated:

Brembs, B., & Wiener, J. (2006). Context and occasion setting in *Drosophila* visual learning. *Learning & Memory*, 13(5), 618-628.

This is a few examples from a fairly brief search of the literature - the authors should make a more comprehensive survey to contextualise their claims.

We have added all the mentioned citations.

Note however that these examples refer to cases of multimodal (contextual) learning, while we only used olfactory stimuli, which constitutes a different case.

C B I
Centre de Biologie
Intégrative - Toulouse

As outlined in the general comments above, it is not a good representation of the extensive literature on configural learning to suggest that current researchers might believe there to be a "strict dichotomy between an elemental and a configural representation of odours" and that it is thus a contribution of this paper to have shown a "continuum" of processing.

Indeed. We removed this sentence and the unique cue (and affiliated explanations) appear explicitly both in introduction and discussion (Lines 54-60, 327-337). However, most of the previous work on the subject assumed a discrete representation of stimuli driving the success (or failure) of animals at Non-elemental tasks. Here we suggest a different interpretation where Elemental, Configural, Unique cue and related processing are actually based on the same perceptual model, which allows animals not only to solve a wide-range of behavioural tasks (requiring either to be very specific or, on the contrary, to show generalization abilities), but also provide flexibility to adapt to the task at hand. Practically, said flexibility originates from the modulation of the initial representation toward either a better discrimination or a better generalization of the stimuli involved, depending on whether the task requires the animal to differentiate them or treat them as the same. Such flexibility has already been described in crustaceans (Livermore et al., 1997), *Drosophila* (Barth et al., 2014, Brembs & Wiener, 2006), *Drosophila* larvae (Chen et al., 2017) and even humans (Howard & Gottfried, 2014).

It is appropriate to consider the MB architecture as a plausible substrate for non-elemental learning. But the discussion could be more mechanistic. There are two ways in which the KC pattern resulting from presentation of two odours together could differ from the sum of each odour individually, even if the activation at the PN level is a simple sum. Firstly, KC seem to act as coincidence detectors for their PN inputs; there may be some KCs that have inputs from PNs that respond to different odours; hence are only activated when both odours are present. Second, if inhibitory feedback limits the total number of active KCs, then even if the initial activation of KCs by PNs for the compound was an exact overlap of the response to the elements, the inhibition would 'prune' this representation to remove any more weakly responding KCs, hence the resulting activation would again represent the compound differently from the sum of the elements. These mechanisms have been explored in some of the modelling papers cited, e.g. 19, and in actual recordings from *Drosophila* KC so deserve clearer presentation,

Indeed, this is exactly what we meant. We clarified the wording of our explanation in the light of your comment. (Lines 296-338). Thanks. The crucial role of inhibitory feedback for non elemental task resolution is particularly engaging as it has also been developed in previous work in honeybee (Devaud et al., 2015, ref 42).

Referee: 2

Comments to the Author(s)

Durrieu et al present the first behavioral evidence that *Drosophila* can learn "non-elemental" conditioning tasks, e.g., A and B are paired with shock when presented individually, but the mixture AB is not paired with shock, thus the fly goes to AB instead of A or B. This is an important finding because it is clearly predicted by our current understanding of the *Drosophila* olfactory system, yet it has never been observed (indeed, a previous study tested for non-elemental learning and did not see it). Thus, this study provides important experimental support for current theoretical understandings of olfactory sensory coding in *Drosophila*. I support publication with minor revisions.

C B I
Centre de Biologie
Intégrative - Toulouse

UNIVERSITÉ
TOULOUSE III
PAUL SABATIER
Université
de Toulouse

1. Discussion: "This result differs from the response of naïve flies, for which a compound is twice as repulsive as the single odorants, at the concentrations used." - Is this derived from the unpaired PI in supplementary figure 1, where untrained flies prefer A over AB? The authors should explain more explicitly how the figure "twice as repulsive" is calculated, or if they aren't confident in the quantitative estimate, soften it to a qualitative "more repulsive."

The reviewer is right, a more cautious and qualitative wording should be used. We have therefore corrected this wording as suggested and clarified this point in our manuscript (Line 120).

2. Discussion: "By modulating its inhibitory activity, the APL neurons could in theory mediate a shift along the generalization / discrimination balance as observed here." I don't think the authors have shown a shift between generalization and discrimination - can they clarify what they mean here, and provide a hypothesis for how APL might be involved?

We agree with the referee, this part deserves clarification. We thus provided a more developed explanation for the so-called Generalization / Discrimination balance, together with a potential underlying mechanism involving APL neurons (Lines 316-336)

3. Can the authors speculate on why, for negative patterning, flies prefer A over AB after 1 training cycle? Why is repetition required to get configural learning? Is it simply that configural learning is difficult? Or would repetitive training shift a balance between elemental vs. non-elemental learning? Is there any theoretical reason to expect such a shift?

Our explanation is presented in the manuscript (L 280-287). We suggest that at first, the AB compound is perceived as an elemental sum of A and B. Thus, A + B, which are innately both aversive, yields more repulsion than A alone. And this is why the task is hard. However over the course of trials, flies learn to treat AB differently than A + B (configural), thus ending up solving the task. This part takes time as in theory initial learning (elemental) is gradually attuned under the influence of APL neurons (see rework lines 307-337). This point is related to the previous one, as we believe that elemental representation admits generalization between compound and components while configural representation is a theoretical situation where the compound is entirely distinct from its constituents.

4. Supplementary figure 1 - can the authors show similar data for the negative patterning experiments as well?

Yes, the supFig1 was modified accordingly.

5. The authors should clarify how AB mixtures were generated - were the two odors at the same concentration as when presented singly (so that an MCH-octanol mixture would have 2.27 mM octanol and 2.62 mM MCH) or were they at half the 'single-odor' concentration? [The latter would complicate the interpretation of the negative featuring experiment - the flies might not be seeing AB as a different 'configuration' from A but rather as a lower concentration of A.] Were the mixtures "liquid" mixtures (i.e. MCH and octanol diluted in the same volume of mineral oil) or mixtures of 2 separate odor-saturated air streams?

We agree that this point needs clarification. The two odours were always presented at the same concentration, whether they were presented singly or in a mixture to avoid the mentioned complication. The compounds were liquid mixtures. We added this information to the M&M part (Lines 108-110).

C B I
Centre de Biologie
Intégrative - Toulouse

2 points that are only suggestions:

6. The authors may wish to frame the paper more around the puzzle that non-elemental learning hasn't been observed even though is clearly predicted from the field's current understanding of olfactory sensory coding in flies (Wessnitzer et al - but even intuitively from first principles - because the KC population response to AB probably includes KCs that didn't respond to either A or B, thanks to normalization and lateral interactions between glomeruli in the antennal lobe). They do mention this problem but I am suggesting to make the tension between experiment and theory more central to the Intro and Abstract. It might make the paper more visible to the sensory coding field (not just the behavior field).

We hesitated indeed to tune our introduction according to the suggested pitch. We chose instead to focus the intro on the elemental/non-elemental literature as our work is based on behavioural experiments. We hope nonetheless that our discussion will make the paper also attractive to the sensory coding field.

7. Given that Wessnitzer et al's model can only learn negative patterning with normalization in the antennal lobe, it may be worth discussing that being able to learn negative patterning might be one functional 'purpose' for having normalization (on top of the usual gain control reasons).

As you said normalisation is a very widespread mechanism. We are not sure we understand what the reviewer means by 'purpose', is it that the need to achieve negative patterning is part of the reason for the evolution of normalisation in the antennal lobe? We tend to think that normalisation in the antennal lobe evolved before the existence of MB. In the absence of proper comparative analysis across species, this claim remains speculative, and thus, we would prefer not to discuss this idea.

Minor points

Discussion: "Even after five training cycles in NP, the flies still exhibited some avoidance of the CS- mixture AB, implying that AB- still carried some of the inhibitory strength associated with its constituents, and thus refutes a purely configural representation (figure 2b, left panel)." - should this be figure 2b, *right* panel?
Typo in 3rd-to-last para: "How such a change in of odour representation could be implemented in the fly's brain?"

Corrected.

Additional notes :

We modified the a) panel of the Fig1 so that it describes all three protocols in a more extensive way.

Appendix B

We would like to start by thanking both the editor and the referees for their time and interest in our topic.

Associate Editor:

Board Member: 1

Comments to Author:

Your manuscript has been re-reviewed by the original reviewers, both of whom agree you have satisfied their previous concerns. One reviewer still has a few minor queries about specific points that would be useful to address or discuss. This should be painless to do and will not require further review. I enjoyed reading this paper and learnt a lot from both it and the reviewer's comments.

I also confirm the data dryad link to the primary data works.

A few typos:

- italicise *Drosophila* in the abstract (line 20,21,26)
- line 57: jointed -> joint ?
- line 86: hyphenate non elemental
- line 217: animal -> an animal
- line 241 preferred significantly -> significantly preferred
- line 282 capitalise and italicise *Drosophila* (and on lines 294, 299, 308)
- line 298/314: should antennal lobes be capitalised?

Done thanks

Reviewer(s)' Comments to Author:

Referee: 2

Comments to the Author(s)

My comments have been fully addressed.

Ref 48 is missing the journal and part of the title but this can be corrected in the proof.

We thank again the referee for his/her comments. We corrected the Ref 48.

Referee: 1

Comments to the Author(s)

This revision has addressed most of my concerns with the original paper. In particular the authors have significantly clarified i) which aspect of their results are original, ii) the mechanistic implications of non-elemental learning, related to known neural circuitry and ii) the generalisability of the results, by repeating the key experiments with alternative odour combinations. Overall, this provides a convincing story that should be of wide interest.

I have the following minor concerns:

lines 173-175 It is not appropriate to conclude that H0 is supported "showing that the amount of experience of flies did not affect the learning of this elemental

discrimination" from a non-significant statistical difference; rather, the experiment did not provide evidence for an effect of amount of experience.

We fully agree with the referee's comment. We rewrote that conclusion adequately (line 230ff).

lines 155-157 Although I accept it is a common procedure in this field, I am still not wholly convinced by the assumption that sets of paired and unpaired groups can be considered true replicates and hence have their scores subtracted to obtain 'corrected PI' scores for analysis. Although the same mean value is obtained by using the 'corrected PI' or the difference of the mean Paired and mean Unpaired, the variance calculation that forms the basis of the t-test is different under these two procedures: what is effectively a paired t-test will give a different result to a two-sample t-test with independent groups. Moreover, it seems very relevant to the discussion of mechanisms that the bias in the 'corrected' PI is sometimes a result of biased behaviour in the unpaired rather than the paired group (see below). As such, the terminology 'corrected PI' seems misleading to me.

Pairs of paired and unpaired groups are no linked randomly, but because the flies are trained simultaneously in the homologous position of their respective experimental set ups. We understand the referee's concern regarding the uncertain link between Paired and Unpaired groups implied by the terminology "Corrected PI". Therefore, we changed it to "Relative PI". We also changed the Supp fig 1 by removing the lines linking Paired, Unpaired and Relative PI points, as it contributes to mislead the reader into thinking these are actually paired data.

I do not fully understand their response to my suggestion, in my review, that the NF learning could be elemental. I did not suggest that the single odour 'A' was attractive, but that it is 'relatively' attractive (i.e. less unattractive) compared to the compound 'AB', and trial repetition makes A more and more aversive (rather than AB less and less aversive) leading to a 'balance' of aversion for the paired group, a remaining imbalance in the unpaired group, and hence a significant change in the 'corrected' PI index. However I accept that this account still implies that AB is processed differently to A+B (i.e. non-elementally) as otherwise AB aversiveness should change in step with the change in aversiveness of A. However, it feels to me that the revision of the paper rather 'buries' the relevant fact that compounds are innately more aversive, which is relevant to understanding the mechanisms.

We thank the referee for the clarification of his/her point. Indeed, we agree with this comment. We added a sentence in the new manuscript (line 177-179) and figure 2 b along with supp fig 1 now show very clearly that the compound is always innately more repulsive than its components.

Supp fig 1, it is good to have included NP data, but I would still like, for completeness, to see the same data for 1 and 5 cycles, as I believe it is important to understand whether or not changes in behaviour are seen during unpaired training in these paradigms.

We agree that for a full account of the presented procedure, 1 cycle and 5 cycles procedures should be displayed in the figure. Thus, we reworked Supp fig 1

accordingly. It is now even clearer that training repetition only improve flies performances in the Paired procedure as t.tests performed between 1 cycle and 5 cycles in the unpaired groups did not show evidence for an effect of repetition there.